# Radiomic Immunophenotyping of GSEA-Assessed Immunophenotypes of Glioblastoma and Its Implications for Prognosis: A Feasibility Study

**DOI:** 10.3390/cancers12103039

**Published:** 2020-10-19

**Authors:** Justin Bo-Kai Hsu, Gilbert Aaron Lee, Tzu-Hao Chang, Shiu-Wen Huang, Nguyen Quoc Khanh Le, Yung-Chieh Chen, Duen-Pang Kuo, Yi-Tien Li, Cheng-Yu Chen

**Affiliations:** 1Department of Medical Research, Taipei Medical University Hospital, Taipei 110, Taiwan; justin.bokai@gmail.com (J.B.-K.H.); 154101@h.tmu.edu.tw (G.A.L.); shiuwen@tmu.edu.tw (S.-W.H.); 2Translational Imaging Research Center, Taipei Medical University Hospital, Taipei 110, Taiwan; s19501062@gm.ym.edu.tw (Y.-C.C.); 194029@h.tmu.edu.tw (D.-P.K.); angela810727@tmu.edu.tw (Y.-T.L.); 3Graduate Institute of Biomedical Informatics, Taipei Medical University, Taipei 110, Taiwan; kevinchang@tmu.edu.tw; 4Clinical Big Data Research Center, Taipei Medical University Hospital, Taipei 110, Taiwan; 5Department of Pharmacology, School of Medicine, College of Medicine, Taipei Medical University, Taipei 110, Taiwan; 6Graduate Institute of Medical Sciences, College of Medicine, Taipei Medical University, Taipei 110, Taiwan; 7Professional Master Program in Artificial Intelligence in Medicine, College of Medicine, Taipei Medical University, Taipei 110, Taiwan; khanhlee@tmu.edu.tw; 8Research Center for Artificial Intelligence in Medicine, Taipei Medical University, Taipei 110, Taiwan; 9Department of Medical Imaging, Taipei Medical University Hospital, Taipei 110, Taiwan; 10Neuroscience Research Center, Taipei Medical University, Taipei 110, Taiwan; 11Department of Radiology, School of Medicine, College of Medicine, Taipei Medical University, Taipei 110, Taiwan

**Keywords:** glioblastoma, radiogenomics, immunophenotypes, first-order statistics, gray-level run length matrix, gray-level co-occurrence matrix

## Abstract

**Simple Summary:**

Characterization of immunophenotypes in GBM is important for therapeutic stratification and helps predict treatment response and prognosis. However, identifying immunophenotypes of patients with GBM requires multiple laboratory experiments and is time consuming. We developed a non-invasive method to evaluate enrichment levels of CTL, aDC, Treg, and MDSC immune cells to classify immunophenotypes of GBM tumor microenvironment with radiomic features of MR imaging. Five immunophenotypes (G1–G5) of GBM can be classified with specific gene set enrichment analysis. G2 had the worst prognosis and comprised highly enriched MDSCs and lowly enriched CTLs. G3 had the best prognosis and comprised lowly enriched MDSCs and Tregs and highly enriched CTLs. Moreover, the developed radiomics models can successfully identified these two groups by immune cell subsets enriched levels prediction. Therefore, it is possible to characterize immunophenotypes of GBM and predict patient prognosis with radiomics methods.

**Abstract:**

Characterization of immunophenotypes in glioblastoma (GBM) is important for therapeutic stratification and helps predict treatment response and prognosis. Radiomics can be used to predict molecular subtypes and gene expression levels. However, whether radiomics aids immunophenotyping prediction is still unknown. In this study, to classify immunophenotypes in patients with GBM, we developed machine learning-based magnetic resonance (MR) radiomic models to evaluate the enrichment levels of four immune subsets: Cytotoxic T lymphocytes (CTLs), activated dendritic cells, regulatory T cells (Tregs), and myeloid-derived suppressor cells (MDSCs). Independent testing data and the leave-one-out cross-validation method were used to evaluate model effectiveness and model performance, respectively. We identified five immunophenotypes (G1 to G5) based on the enrichment level for the four immune subsets. G2 had the worst prognosis and comprised highly enriched MDSCs and lowly enriched CTLs. G3 had the best prognosis and comprised lowly enriched MDSCs and Tregs and highly enriched CTLs. The average accuracy of T1-weighted contrasted MR radiomics models of the enrichment level for the four immune subsets reached 79% and predicted G2, G3, and the “immune-cold” phenotype (G1) according to our radiomics models. Our radiomic immunophenotyping models feasibly characterize the immunophenotypes of GBM and can predict patient prognosis.

## 1. Introduction

Glioblastoma (GBM) is the most common malignant brain tumor in the central nervous system. Despite the availability of standard treatment, such as tumor resection followed by radiotherapy and temozolomide (TMZ) [1], the median survival time of patients is only 14 months, and recurrence is always [2]. An increasing number of studies have also attempted to identify distinct immune responses within the tumor microenvironment (TME), considering these responses to be intervention clues as to assist immune system to defend against tumor cells [3,4,5].

Immunophenotyping is the identification of the abundances of subpopulations of immune cells to estimate immune response [6,7]. The increased frequency of cytotoxic T lymphocyte (CTL) infiltration in the GBM microenvironment has been demonstrated to be capable of improving patients’ survival by specifically targeting and killing invading tumor cells [8,9,10,11,12,13]. Moreover, CTLs must be induced by activated dendritic cells (aDCs) [14], which are specialized antigen-presenting cells (APCs) for clearing tumor cells. In contrast to CTLs and aDCs, myeloid-derived suppressor cells (MDSCs) and regulatory T cells (Tregs) are two major immune subsets that cause immune suppressive effects in the TME [15], and they also play important roles in T-cell exhaustion [16]. Both MDSC and Treg subsets are capable of suppressing T-cell activation and IFN-γ production, and they are correlated with poor patient prognosis [17,18]. The dynamic of immune subsets has, since recently, been considered to be a biomarker and predictor for the response rate to cancer treatment [19]. Therefore, the efficient prediction of immune dynamics constitutes an unmet medical need with specific respect to improving treatment efficacy.

Due to intratumoral heterogeneity within different regions of individual tumors, tumor biopsy may suffer under-sampling when representing the pathological molecular profile in a whole tumor [20,21]. Magnetic resonance imaging (MRI) is a noninvasive approach for characterizing the anatomical, functional, microstructural, and metabolic properties of tissues. In MRI, different image contrasts and imaging sequences are used to reveal tumor heterogeneity in an individual tumor. Radiomics refers to the extraction of many high-dimensional quantitative image features that describe the intensity, texture, and geometrical characteristics indicated by a given set of tumor radiographic data [22]. These features can be utilized to build predictive models for identifying the molecular subtype and immune-related gene expression level within the TME [23,24,25,26]. In addition, several approaches have been demonstrated as being capable of identifying intratumoral immunophenotypes; these approaches include flow cytometry [27,28], immunohistochemistry [29], and the transcriptome expression profiling of distinct metagene lists by using gene set enrichment analysis (GSEA) [30,31,32,33,34,35,36]. These approaches can be used to efficiently estimate the abundances of immune subsets, including CTLs, aDCs, Tregs, and MDSCs. To the best of our knowledge, studies have yet to utilize radiomic features to construct models for identifying immunophenotypes in GBM. Thus, in this study, we formulated machine learning-based radiomic models that use radiomic MRI features to distinguish various intratumoral immunophenotypes in GBM. We employed radiomic features that were derived from various MR images to investigate possible immune phenotypes and predict prognosis.

## 2. Materials and Methods

### 2.1. GBM Cohorts

To obtain reliable results, genomic datasets from two types of platforms (microarray and next-generation sequencing) were used in this study. We downloaded GBM RNASeq (level 1) data (*n* = 154) from TCGA at https://www.cancer.gov/tcga through the Genomic Data Commons (GDC) data transfer tool; we were authorized by the Electronic Research Administration (eRA) Commons and the Database of Genotypes and Phenotypes (dbGaP) to access this level 1 data. In addition to the next generation sequencing dataset, we downloaded GBM microarray data (*n* = 564) from the International Cancer Genome Consortium (ICGC) data portal [37], the platform used was the Agilent 244K Custom Gene Expression (G4502A-07-2). The MR imaging data of 116 patients with GBM were obtained from The Cancer Imaging Archive (TCIA) database. This study was approved by TMU-Joint Institutional Review Board (code: N201603086).

After obtaining the summary statistics of the datasets, we noted that 32 and 111 MR imaging samples had their own RNASeq and microarray gene expression data, respectively. Because of the probe-set limitation of microarrays in detecting several genes within specific gene sets—which were introduced in the section on the estimation of the enrichment level of immune subsets—the array data were not used to build models but rather used as adjuvant data to determine the signature correlations between radiomic feature and expressed gene. The imaging samples without using in the training set were used as an independent testing set. Additionally, for the prognosis evaluation of patients with GBM in this study, we only focused on the patients with wild-type IDH and used survival days and vital status as evaluation indicators.

### 2.2. Genomics Data Preprocessing

The TCGA level 1 (RNASeq) data used in this study were processed through a series of steps. Two major processes were employed: Reads alignment and gene/isoform expression quantity estimation. For these two processes, we employed HISAT2 [38] and StringTie [39,40], respectively. The expression level utilized the fragments per kilobase of transcripts per million mapped reads (FPKM) [41] to determine the gene expression level. The formula for FPKM is as follows:(1)$$\mathrm{FPKM}\;=\;\frac{\mathrm{total}\;\mathrm{fragments}}{\mathrm{mapped}\;\mathrm{reads}\;\left(\mathrm{millions}\right)\times \mathrm{exon}\;\mathrm{length}\;\left(\mathrm{kilobase}\;\mathrm{pair}\right)}$$

The GBM dataset of the Agilent microarray platform and the intensities of genes were normalized using the locally weighted scatterplot smoothing (LOWESS) normalization method [42].

### 2.3. Enrichment Level Estimation of Immune Subsets

The major immune subsets of (1) CTLs, (2) aDCs, (3) Tregs, and (4) MDSCs were used to explore immune system-related differences within the TME among patients with GBM. The enrichment levels of these four subsets within the TME could be manipulated with the expressions of distinct multiple genes. The candidate metagene lists of aDCs, Tregs, and MDSCs were detailed in a previous study [33], and that of CTLs was detailed in another previous study [32]. The abundances of immune cell subsets within the TME could be estimated using normalized enrichment scores in GSEA [36]. Consequently, 5, 51, 26, and 58 genes could be used to evaluate the enrichment levels of CTL, aDC, Treg, and MDSC immune subsets, respectively. The metagene lists and statistics of the used genes among various gene expression platforms were shown in Appendix A. In addition, the trends of enrichment levels in the four immune subsets from low to high were scaled with immunogram scores (IGSs) (ranging from 0 to 5) and plotted in a radar chart [33].

### 2.4. Imaging Processing and Radiomic Feature Extraction

We established a postprocessing protocol on MR images to reduce the discrepancy between imaging parameters employed by different hospitals [43]. We adjusted the image resolution to have a voxel size of 0.75 × 0.75 × 3.00 mm^3^ without gaps between consecutive slices for each MR modality. ADC maps derived from diffusion-weighted imaging (DWI) were then registered to T1C images by a six-parameter rigid body transformation and a mutual information algorithm. MRI intensity was normalized into standardized ranges for each imaging modality. The region of interest (ROI) covering the total tumor regions were defined by two experienced neurosurgeons and neuroradiology researchers. T1C images and ADC maps were used for the extraction of radiomic features. The wavelet coefficients of each MR image were also calculated per a method described in a previous study [43].

### 2.5. Machine Learning Method and Variable Selection

The enrichment of immune subsets in the TME can be evaluated through the expression of genes involved in specific gene sets. To survey the effects of radiomic features on classifying enrichment levels of the four aforementioned immune cells, we implemented logistic regression. The regression model had radiomic features as predictors and the immune cell subset enriched level (0: Low, 1: High) as the dependent variable. Because the data set was small, leave-one-out cross-validation was used as an evaluation method to improve the reliability of performance results.

Furthermore, to obtain those radiomic features in MR imaging data that were highly associated with the enrichment levels of immune cell subsets, we implemented three strict procedures to select candidate features for model construction. The first step was filtering the common signatures of features among patients: Features were removed while the values among larger than 90% of patients were missing or zero. Because too many missing data points and zero values of each feature in the data set of patients, it was difficult to exactly determine the association between features and enriched levels of immune cell subsets. Thereafter, to extract features with a stable correlation between radiomic features and the gene signatures of various platforms, we implemented the second step: We filtered features with reliable signatures, which indicated the consistency of the correlation between expressed profiles from radiomic features and the genes involved in the gene sets of four subsets under different platforms—such as RNASeq and the Agilent microarray. For instance, the profiling of radiomic features to the expression of genes had a correlation among patients not only in RNASeq but also in microarray data. This correlation was positive for the RNASeq data and also positive for the microarray data. Subsequently, feature filtering in the third step was implemented with two types of algorithms: Random forest and information gain. The filtering algorithm of random forest, the function *Importance* can be used to evaluate the effectiveness of each feature when the random forest model was constructed with the *randomForest* function in R’s *randomForest* package. For each feature, the value of the mean decrease of Gini was obtained from the *Importance* function analysis. Because a higher value indicates a more important variable, the value of each feature must be at least larger than zero. In addition, we used the entropy-based function *information.gain* to estimate weights of features based on the correlation between predictors and the dependent variable. Similarly, the value of feature weights must be larger than zero and ranked in the descending order. Hence, the top-rank features can be selected according to the cutoff value; here, the default cutoff value was set as the mean of weights. However, if no features met the criteria, the value would be set as the median of weights.

## 3. Results

### 3.1. Patient Characteristics

In this study, all GBM samples (RNASeq, *n* = 154) were used to explore four immune cell subsets enriched profiles and to realize whether existed specific enrichment patterns among them. The characteristics of these patients were summarized in Appendix A. On the other hand, 32 samples of imaging data with RNASeq data were utilized as a training data set, and another 84 samples without RNASeq data were utilized as an independent testing set. The characteristics of patients from whom these data were obtained are presented in Table 1.

### 3.2. Identification of Immunophenotypes Based on Four Immune Cell Subsets

Based on the different enrichment levels of four immune subsets (CTLs, aDCs, Tregs, and MDSCs), patients with GBM can be unsupervised clustering into five immunophenotype groups G1 to G5 (Figure 1), namely immune-cold phenotype (almost all immune subsets are lowly enriched) (G1), lowly enriched CTLs and highly enriched MDSCs (G2), only highly enriched CTLs (G3), highly enriched CTLs and highly/lowly enriched in others (G4), and immune-hot phenotype (almost all subsets were highly enriched) (G5). Among patients with GBM, more than half the proportion of immunophenotypes belonged to G5 (26.6%) or G1 (37.7%). The proportion of G3 was the lowest (6.5%) in the study cohort. Table 2 summarizes the proportions of patients in each immunophenotype group. The immunogram scores of the four immune subsets of patients were compared within and between groups (Figure 2a).

### 3.3. Survival Prognosis Evaluation of Patients with Various Immunophenotypes

The prognosis of the classified GBM was analyzed in terms of their survival information by using the Kaplan–Meier method (Figure 2b). The survival curve of these five groups of patients having the various immunophenotypes indicated that patients in the G3 group exhibited the best prognosis, and 50% of patients could survive more than 500 days (median: 867 days). Patients in the G2 group exhibited the worst prognosis, and <50% of patients could survive more than 270 days (median: 266 days). The median overall survival (mOS) duration of the remaining groups was 408, 330, and 357 days for G1, G4, and G5, respectively. The pair-wised log-rank test for the curves demonstrated that G2 and G3 achieved statistical significance (*p* = 0.018). These results indicated that patients with a high CTL level exhibited better survival outcomes than did patients with high MDSC and low CTL infiltration.

### 3.4. Performance Evaluation of Models for Enrichment Level of Immune Cell Subsets

To construct models with suitable radiomic features, a series of filtering criteria were used to analyze these 9809 total features (Figure 3). First, we obtained 8223 contrast-enhanced T1 (T1C) and 8178 apparent-diffusion-coefficient (ADC) radiomic features after removing features with too many missing and zero values among patients. Second, we obtained 4872 T1C and 2534 ADC stable features after further filtering based on the consistent correlation between features and genes across different platforms. Finally, based on two feature selection algorithms, we extracted 9, 5, 5, and 18 T1C features and 6, 20, 2, and 11 ADC features as predictors for constructing the CTL, aDC, Treg, and MDSC subset models, respectively. Subsequently, the performance of each model was evaluated by leave-one-out cross-validation. For T1C imaging data, the average accuracies of the CTL, aDC, Treg, and MDSC models were 0.72, 0.75, 0.81, and 0.88, respectively; for ADC imaging data, the average accuracies of the aforementioned models were 0.71, 0.61, 0.68, and 0.79, respectively. The areas under the curves (AUCs) of T1C-trained and ADC-trained models are also summarized in Figure 3. According to the performance evaluation, T1C features yielded better distinguishability of the enrichment levels of all immune cell subsets relative to ADC features. Subsequently, we used independent testing data (*n* = 84) to evaluate the effectiveness of T1C-trained models. We could classify 19, 16, 17, 23, and six patients into G1-, G2-, G3-, G4-, and G5-prediction groups, respectively. The prognosis analysis of the G3-prediction and G2-prediction groups indicated the highest and lowest mOS of >500 days and <270 days, respectively, among the five groups. The log-rank test for survival curves indicated that G3-predicted patient’s groups have better survival than G2-predicted patient’s groups (*p* = 0.019, as shown in Figure 4). The G1-prediction and G4-prediction groups had mOS of 388 and 424 days, respectively. Moreover, the pairwise comparisons of the predicted groups and observed groups indicated that the trends and mOS days of the G1-prediction survival curve closely accorded with the G1 curve for the observation data (Appendix A). In addition to the survival prognosis, parts of testing data have the corresponding Agilent microarray data to identify their immunophenotypes in predictive G2 and G3 groups. However, several genes within gene sets can be detected in RNASeq data, but cannot be detected in the microarray. Thus, we first test whether the modified gene sets have the similar trends of enrichment levels to G2 and G3 groups to the original gene sets from RNASeq data (Appendix A). The trends of immune cell subsets enriched levels between G2 and G3 could demonstrate that the modified gene sets (remove several genes) were capable to speculate potential immunophenotypes within G2 and G3 groups. Therefore, we applied the modified gene sets to evaluate CTL, aDC, Treg, and MDSC enriched levels in predicted G2 and G3 groups by using microarray data. The trends of four immune cell enriched levels between G2-prediction and G3-prediction has similar trends with G2 and G3 (Figure 5 and Appendix A). These results jointly indicate that the radiomic-based immunophenotyping models could distinguish patient groups with various prognoses in GBM.

### 3.5. Effectiveness of Radiomic Features for Trained Models

Initially, the total signatures of radiomics features were generated from four feature classes: First-order statistics, gray-level run length matrix (GLRLM), gray-level co-occurrence matrix (GLCM), and shape and size. The proportions of each feature class are shown at the top of Figure 6: first-order statistics constituted the major feature type (97% of total radiomic features), and GLCM, GLRLM, and size and shape constituted the minor types (0.1% to 2%). Subsequently, these radiomic features were subject to association analysis between features and enriched levels of immune cell subsets; critical features were further selected from different algorithms, and the proportions of feature classes within the trained models from T1C and ADC MR imaging differed. For the T1C-trained models, 65% were first-order statistics, and 35% were GLRLM radiomic features; for the ADC-trained models, 92% were first-order statistics, 5% were GLRLM radiomic features, and 3% were GLCM radiomic features (bottom of Figure 6). Appendix A provide further details pertaining to feature name, wavelet transform, feature class, importance ratio within distinct models, and statistical test results of essential radiomic features derived from T1C and ADC. In total, 13 distinct first-order statistics and five texture features prior to SIFT algorithm and wavelet transform were further summarized (Table 3). Six and five features derived from T1C and ADC, respectively, exhibited imaging-sequence specificity. For example, “Median,” “Entropy,” “Skewness,” “Third quartile,” long run high gray-level emphasis (LRHGLE), and short run low gray-level emphasis (SRLGLE) features were specifically utilized in T1C-trained models. By contrast, “Maximum,” “Range,” “Uniformity,” short run high gray-level emphasis (SRHGLE), and IMC1 features were only used in ADC-trained models. Figure 7 presents the statistical significance and importance ratio of these radiomic features for the models. These findings demonstrate the effectiveness of the features in model prediction. Features derived from T1C MR imaging (e.g., “LLL_SIFT_Median” and “LLH_SIFT_Mean absolute deviation”) for the CTL model had an explanatory power of 61%; “HLH_LRHGLE,” “LHL_SRLGLE,” and “HHL_SRLGLE” for the aDC model had an explanatory power of 59%; “HHH_SRE” and “HHL_SRLGLE” for the Treg model had an explanatory power of 45%; “LLL_SIFT_Third quartile,” “HHH_SRE,” and “HLH_SIFT_Root mean square” for the MDSC model had an explanatory power of 43%. Features derived from ADC MR imaging (e.g., “HHL_SRHGLE,” “SIFT_Variance,” and “SIFT_Energy”) for the CTL model had an explanatory power of 66%; “LLL_SIFT_Variance,” “SIFT_Variance,” “HHL_SIFT_Maximum,” and “SIFT_Energy” for the aDC model had an explanatory power of 39%; “HHL_SIFT_Range,” “HHH_SIFT_Mean absolute deviation,” “HHH_SIFT_Energy,” and “HHH_SIFT_Mean” for the MDSC model had a 11% explanatory power. All these features for all models reached statistical significance.

Observation of characteristics of radiomic features used in the models present that larger than 75% of these derived from two T1C and ADC imaging sequences are transformed through wavelet transformations, and those from T1C were even as high as 97% (Appendix A). Moreover, the proportions of transformed high (LLH-HHH) and low (LLL) frequency bands for features derived from different imaging sequences varied widely (89% and 8% in T1C; 55% and 20% in ADC).

## 4. Discussion

Immunotherapies for tumors are considered to be effective; however, immune heterogeneity means that GBM remains largely refractory [44]. Because treatment causes dynamic changes in the immune population, a noninvasive method for identifying immunophenotypes benefits patients with respect to the selection of a suitable immunotherapy method. In this study, we discovered that patients with GBM can be classified into five distinct groups and that two groups (G2 and G3) significantly differed with respect to survival prognosis. We also constructed machine learning-based radiomic models to classify levels of four infiltrating immune subsets (CTLs, aDCs, Tregs, and MDSCs), and the models can predict patient prognosis by using independent testing data. This study is the first to use noninvasive MR imaging data to identify immunophenotypes among patients with GBM. The model performance and independent data testing demonstrated that our model feasibly elucidated the TME by using the radiomic features of MR imaging data.

Immunohistochemistry-based methods have recently been used to distinguish between the immune phenotypes of different tumors into cold (immune desert), altered-excluded (immune cell excluded), altered-immunosuppressive (immune cell recruitment and expansion limited), and hot (inflamed) types; these types are based on two lymphocyte populations (CD3 and CD8) at the infiltrated level and the spatial distribution of the TME [45]. Additionally, a few trials on patients with GBM have demonstrated a positive correlation between infiltration in the CTL subset and favorable prognosis [9]. In this study, we utilized the GSEA method, which is based on the expression profiles of multiple genes, to evaluate the high and low enrichment levels of CTLs among patients with GBM (The Cancer Genome Atlas [TCGA]-GBM, *n* = 154) to, in turn, simulate the inflamed and immune desert phenotypes of tumors. However, the survival curves of both types of tumors did not significantly differ (Appendix A). Moreover, when classifying GBM patients based on the enrichment level of the four immune cell subsets, we noted that the mOS days of these distinct groups (G1 to G5) potentially differed, and the survival curves of G2 and G3 even significantly differed. This demonstrated that the estimation of infiltration in multiple immune cells in relation to the immune phenotypes of GBM aid the prediction of patients’ survival prognosis.

In addition to classifying GBM patients with distinct immunophenotypes based on a GSEA analysis of genes associated with CTLs, aDCs, Tregs, and MDSCs, we also constructed noninvasive radiomics models to predict these immunophenotypes by using MR imaging (T1C and ADC sequences) features. Our models also aid the prediction of not only potential immune responses to tumor cells but also survival prognosis. Although the developed models use more reliable radiomic features, which we filtered through various criteria, the performance levels of the T1C-trained models were more acceptable than that of the ADC-trained model. 

According to the evaluation performed using independent test data, the developed radiomics models of our study could identify three groups (G1, G2, and G3) reliably and aid both the prediction of prognosis and the selection of suitable treatment strategies. For the prediction of prognosis, G3 had the most favorable prognosis among the groups, which was attributable to the highly enriched CTL subset against tumor cells not being exhausted by Treg and MDSC subsets [9]; G2 had the worst prognosis, which was attributable to the highly enriched level of MDSCs but the lowly enriched levels of CTLs for suppressing the efficacy of tumor cell clearance [46]. With respect to patient treatment strategies, our radiomic immunophenotyping model could predict patients who had highly enriched MDSCs (G2 group), and the model suggested for them to receive therapy that can eliminate or inactivate MDSCs [47]. Our model could also predict patients who had low enrichment levels of CTL, aDC, Treg, and MDSC subsets (G1 group), which are devoid of T cells and considered as less conducive to immunotherapies [48]. Among these patients, antitumor immunity can be improved using the dendritic cell vaccine (DCV) method, which loads specific antigens to promote a tumor-specific immune response (such as CTL) against tumor cells [48,49].

T1C-trained models, which were based on the SIFT algorithm and wavelet transform, had several specific and significant radiomic features that could explain the enrichment levels of CTL, aDC, Treg, and MDSC in a GBM microenvironment. The values of the features of “Median” and “Mean absolute deviation” were positively and negatively correlated with CTL enrichment levels, respectively. We attribute this finding to the high median intensity but low intensity deviation of tumor voxels, which can potentially represent the highly enriched CTL subset in the TME. The magnitudes of the two intensity-related features “Third quartile” and “Root mean square” were positively and negatively related to the level of MDSCs, respectively. Thus, a higher enrichment level of MDSCs entails higher “third quartile” and lower “root mean square” intensity values within the tumor region, respectively. In addition, the value of the textural feature “SRE,” which is a measure of the distribution of short run lengths, was negatively correlated with run length, positively correlated with the fineness of textures, and negatively correlated with the level of MDSCs. Therefore, the tumor imaging of highly enriched MDSCs may present a less fine texture. For the enrichment levels of aDC and Treg subsets, the features used in the models were those relating to texture types. The two textural features of “LRHGLE” and “SRLGLE” were positively and negatively correlated with the enriched level of aDCs, respectively. “LRHGLE” measures the distribution of long run lengths and high gray-level values, with a higher value indicating coarser structural textures and a greater concentration of high gray-level values in the image. “SRLGLE” measures the distribution of short run lengths and low gray-level values, with a higher value indicating finer structural textures and a greater concentration of low gray-level values in the image. Based on both of “LRHGLE” and “SRLGLE” definitions, although tumors in T1C imaging have coarser structural textures and a greater concentration of high gray-level values, the aDC subset is potentially enriched at high levels. The values of the features “SRLGLE” and “SRE” were positively and negatively correlated with an enriched level of the Treg subset, respectively. Additionally, “SRE” is more important (larger coefficient values) than “SRLGLE” is in the Treg model. Therefore, although Tregs are highly infiltrated within the TME, the characteristics of the T1C imaging of tumor voxels feature a less fine texture and are similar to the imaging properties of the enrichment levels of the MDSC subset in GBM.

Despite the usefulness of the developed radiomics immunophenotyping models, our study has some limitations. One of the concerns is sample size limitation; only 32 samples contained gene expression data and its corresponding MR imaging data from TCGA and TCIA database, respectively. However, we use independent testing data (*n* = 84), which comprised MR imaging data and clinical information to validate the effectiveness of radiomics immunophenotyping models that can predict patients’ survival. Moreover, distinct platforms—such as microarrays, which have comprehensive probesets for conducting GSEA to estimate the enrichment levels of immune cell subsets—are required in future studies, considering the usefulness of further verifying the effectiveness of radiomic features in the constructed models.

## 5. Conclusions

The results demonstrate that machine learning models can utilize radiomic features to predict the enrichment levels of various immune subsets within the TME; these predictions can, in turn, be used to differentiate between various immunophenotypes present in patients with GBM. The results also demonstrate that specific radiomic features can be feasibly used to detail the best and the worst prognosis of immune phenotypes, which were correlated with the enrichment levels of MDSC and CTL subsets, as well as the “immune-cold” phenotype. Therefore, radiomic immunophenotying has promise as a tool for predicting patient prognosis and putative tumor progression, potentially even providing a basis for flexible treatment strategies.

## Figures and Tables

**Figure 1 cancers-12-03039-f001:**
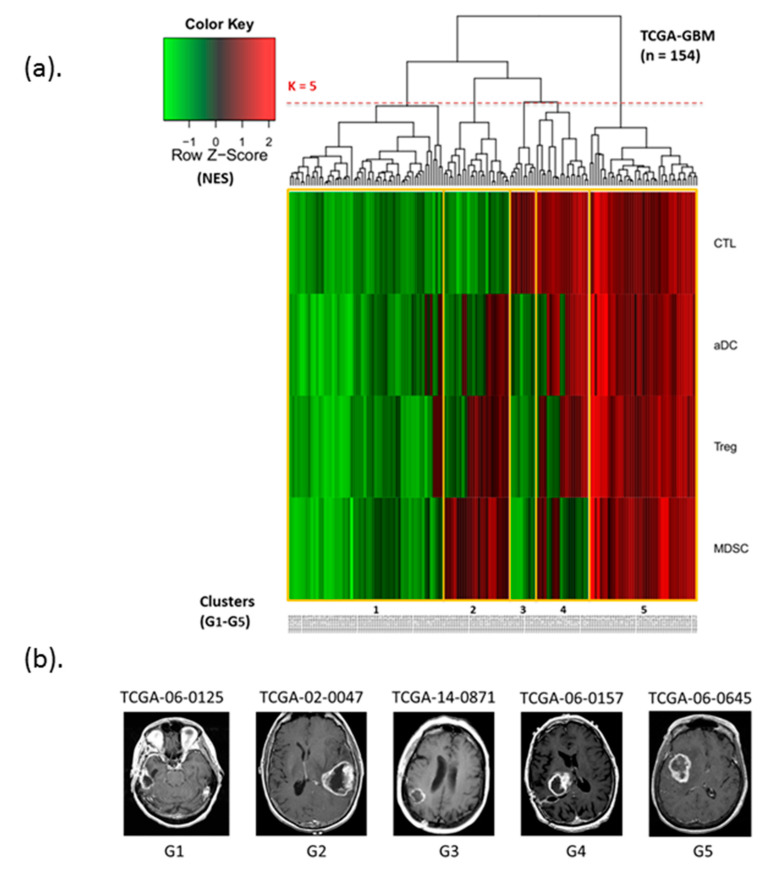
Characterization of glioblastoma (GBM) immunophenotypes by four immune subsets. (**a**) Enrichment levels of immune cell subsets from low to high are represented by a color gradient from green to red. Unsupervised clustering was based on the enrichment profiles of cytotoxic T lymphocytes (CTL), activated dendritic cells (aDC), Treg, and myeloid-derived suppressor cells (MDSC) subsets; patients with GBM could be divided into five subgroups (*K* = 5). (**b**) Imaging data (T1 contrast imaging) from each five subgroups were also listed.

**Figure 2 cancers-12-03039-f002:**
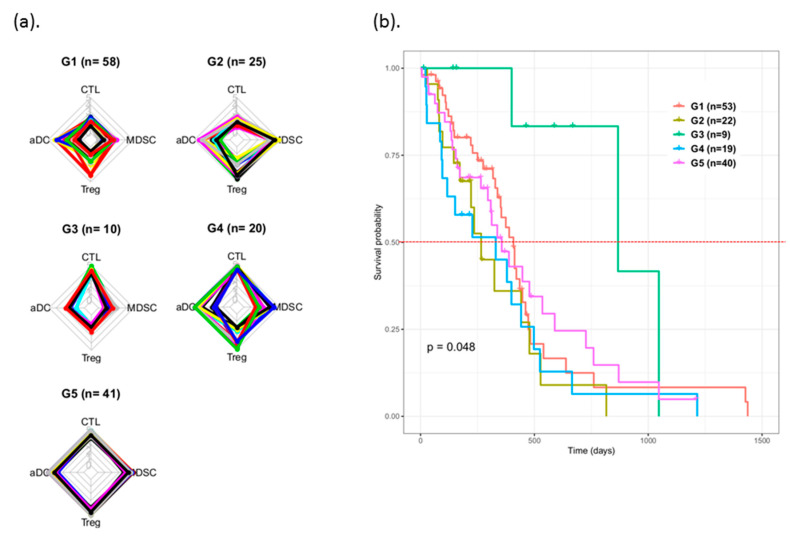
Prognosis of five immunophenotypes of patients with GBM. (**a**) Enrichment levels of CTL, aDC, Treg, and MDSC subsets were evaluated in terms of the immunogram score (IGS) (ranging from 0 to 5) and plotted in a radar chart. Each subset formed individual axes that have been arranged radially around a point. The IGS of each immune cell subsets are depicted by the node on the distinct axis. Each patient is represented by their own line that connects data values from each axis. (**b**) Survival curves of patients within five groups with different immune phenotypes; curves were statistically significant after a log-rank test (global *p* < 0.05). Each median overall survival times of curves is represented by a dotted red line.

**Figure 3 cancers-12-03039-f003:**
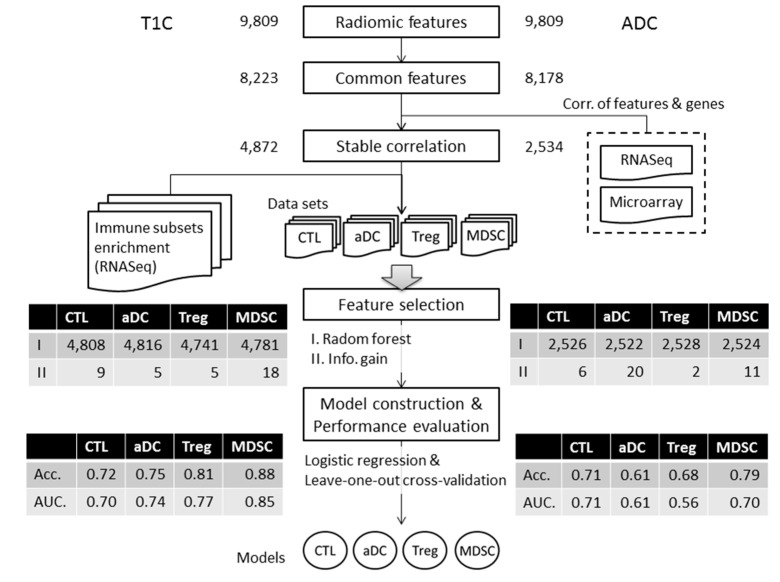
Radiomic-feature filtering for the model construction of four immune cell subsets. Several criteria for filtering the radiomic features of T1 contract enhancement (T1C) and apparent diffusion coefficient (ADC). Numbers of features decreased step by step. Thus, various important features from T1C and ADC for different immune subsets could be used for model construction. Model performance was evaluated with leave-one-out cross-validation.

**Figure 4 cancers-12-03039-f004:**
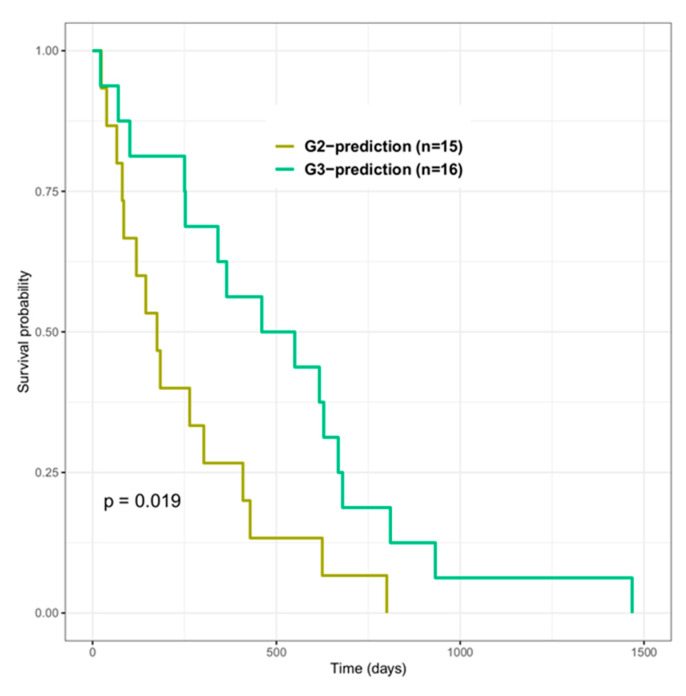
Effectiveness of models for T1C with G2- and G3-prediction groups. G2- and G3-prediction groups were determined using independent testing data (*n* = 84) fed to the T1C radiomics models. The log-rank test for the survival curves between the two groups indicated statistical significance.

**Figure 5 cancers-12-03039-f005:**
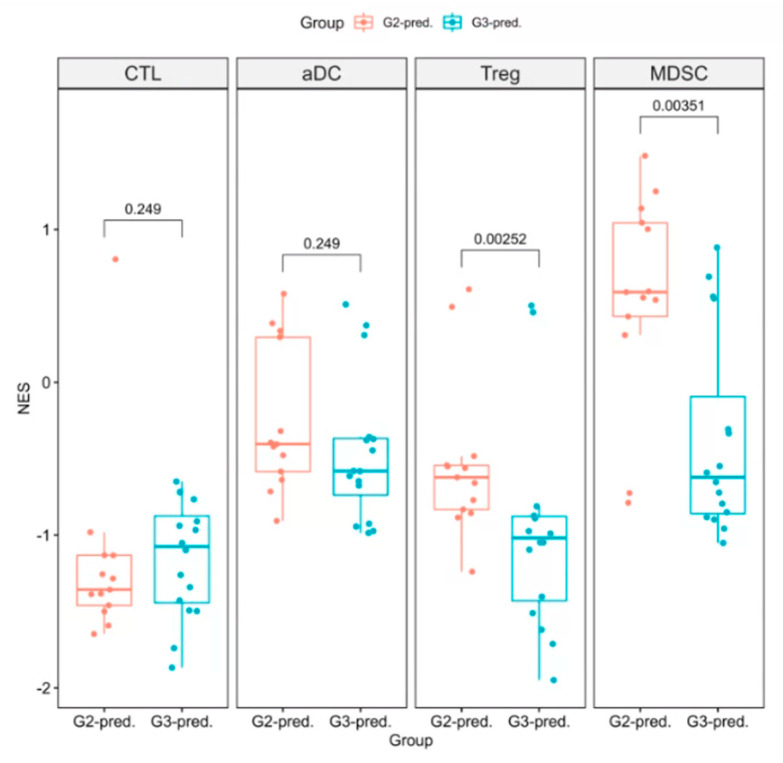
Enrichment levels of immune cell subsets between G2- and G3-prediction groups with microarray data. G2- and G3-prediction groups were determined using independent testing data (*n* = 84) fed to the T1C radiomics models. X-axis indicates groups with different immune phenotypes, which represented in Figure 1. Y-axis indicates enrichment levels of immune cell subsets with (NES). The enrichment levels of aDC, Treg, and MDSC in G2-prediction group were higher than G3-prediction group, and Treg and MDSC even reached statistical significance (Wilcoxon test). Only CTL immune cell subsets were enriched higher in G3-prediction group than in G2-prediction group.

**Figure 6 cancers-12-03039-f006:**
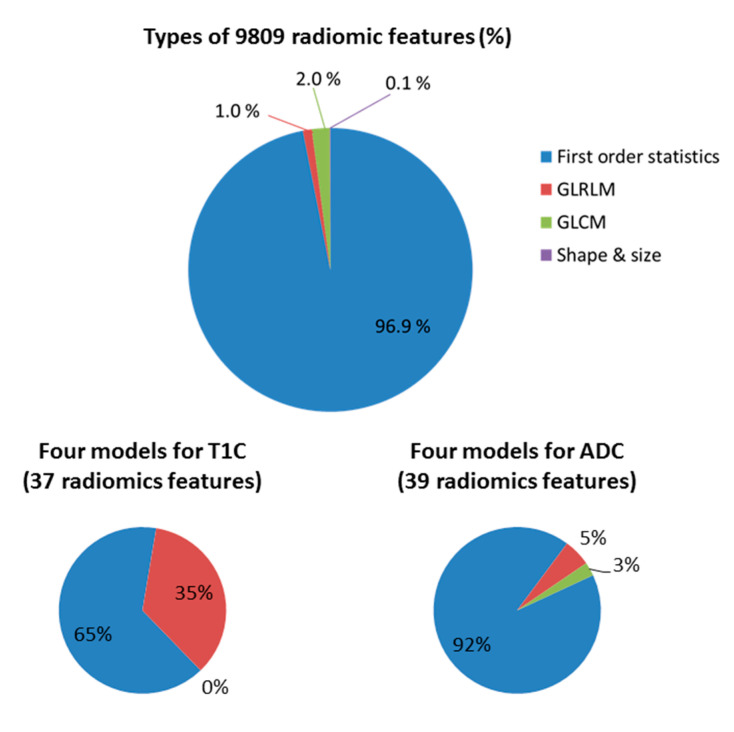
Proportions of feature classes before and after filtering. Four classes of radiomic features, including first-order statistics, gray-level run length matrix (GLRLM), gray-level co-occurrence matrix (GLCM), and shape and size, were used in this study. The top figure indicates the proportions of each class before feature filtering. The bottom left figure indicates the percentages of feature classes used for models trained using T1C magnetic resonance (MR) imaging data. Similarly, the bottom right figure indicates the percentages of feature classes used for models trained using ADC MR imaging data.

**Figure 7 cancers-12-03039-f007:**
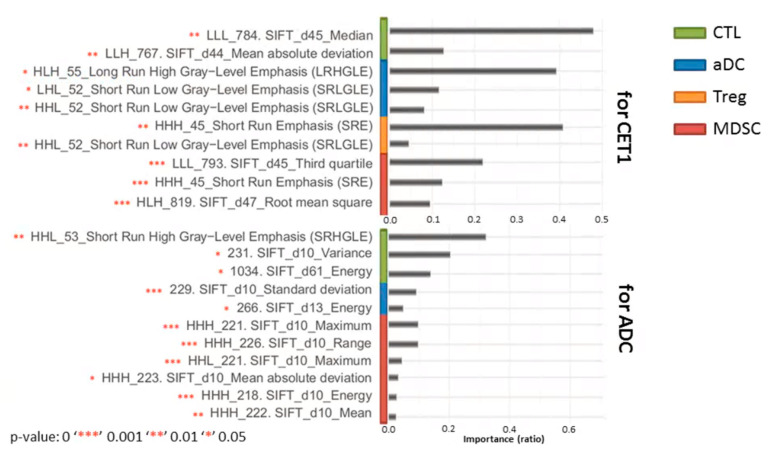
Statistical significance of radiomic features for different models. A chi-squared test was used to evaluate the importance of features for models. Number of stars indicates the extent of statistical significance had by features for each model. (For instance, one star “*” means *p* value < 0.05 and so on.) Otherwise, the importance ratio of features represented the features’ effectiveness relative to other features in the same models.

**Table 1 cancers-12-03039-t001:** Demographic characteristics of patients whose data were used in the training and testing data sets.

Datasets	Training Data (*n* = 32)	Testing Data (*n* = 84)
**Data**		
RNASeq	Yes	No
MR images	Yes	Yes
**Gender**		
Male	20	50
Female	12	34
**Age**		
Median (Q1–Q3)	62.5 (52.3–70.3)	59 (52.0–66.0)
**Survival days**		
Median survival (Q1–Q3)	332.7 (170.0–548.8)	332.2 (179.1–560.3)
**IDH1/2** *		
Wild-type	29	79
Mutant	3	5

* IDH1/2: Isocitrate dehydrogenase 1 or 2.

**Table 2 cancers-12-03039-t002:** Proportion of patients following unsupervised clustering with normalized enrichment scores.

Cluster	Patients (n)	Ratio (%)
G1	58	37.7
G2	25	16.2
G3	10	6.5
G4	20	13.0
G5	41	26.6

**Table 3 cancers-12-03039-t003:** Summary of distinct radiomic features for models derived from T1C and ADC.

Feature Name	Feature Class	T1C	ADC
Energy	First-order statistics	√	√
Entropy	First-order statistics	√	
Maximum	First-order statistics		√
Mean	First-order statistics	√	√
Mean absolute deviation	First-order statistics	√	√
Median	First-order statistics	√	
Range	First-order statistics		√
Root mean square	First-order statistics	√	√
Skewness	First-order statistics	√	
Standard deviation	First-order statistics	√	√
Third quartile	First-order statistics	√	
Uniformity	First-order statistics		√
Variance	First-order statistics	√	√
LRHGLE *	GLRLM	√	
SRE *	GLRLM	√	√
SRHGLE *	GLRLM		√
SRLGLE *	GLRLM	√	
IMC1 *	GLCM		√

* LRHGLE: Long run high gray-level emphasis; SRLGLE: Short run low gray-level emphasis; SRE: Short run emphasis; SRHGLE: Short run high gray-level emphasis; IMC1: Informational measure of correlation 1

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
