# Peer review of "Radiomic Immunophenotyping of GSEA-Assessed Immunophenotypes of Glioblastoma and Its Implications for Prognosis: A Feasibility Study"

_cancers, 2020, doi:10.3390/cancers12103039_

Round 1

Reviewer 1 Report

In this paper, the authors present their work first identifying different gene signatures corresponding to different immune-phenotypes in the tumor microenvironment and then their work creating a radiomics model to predict, globally, which of 5 immune-phenotypes an individual patient’s tumor should be categorized. Overall, the topic of the paper is interesting and it seems that the methods are sound. However, there are a few additional details that need to be incorporated and I think the authors should consider reorganizing the paper for better clarity. I also think the emphasis in the paper is not fully reflected in the title and the impact of the results may be overstated. Let me give more details on each of these points.

Regarding the organization, the sections currently go: intro, results, discussion, methods, conclusion. This organization can work, however, I find in radiomics papers the methods are very critical to understanding the results and so I would encourage the authors to put the methods in front of the results. Even if they choose not to, there are still some details missing. Within the results section, the “patient characteristics” section makes no mention of the TCGA data and the table also ignores this patient set, however the very next two sections are entirely based on TCGA data. This needs to be addressed as otherwise it is very confusing as the numbers don’t add up. An additional comment in the methods section would also be helpful regarding why the authors chose to focus on T1C and ADC only. The patients in the TCIA have many more sequences. Also, why did the authors choose to keep the features of the images separate?

Regarding the emphasis in the paper, it seems to me that the title is completely focused on the radiomics results. However, the discovery and definition of the gene sets through GSEA also seems of great interest. I would encourage the authors to tweak their title to reflect this. Also, I think it is necessary for them to publish the specific gene sets that they used in the supplementary material.

Finally, while the authors admit in their title that this is a feasibility study, the criteria they use for validation with the testing set is not very strong. They merely confirm that the immune-phenotypes demonstrate similar, relative survival curve patterns as the training data. (i.e. G3 has the longest survival) While this is suggestive of similar tumor microenvironments, it certainly does not imply that the microenvironment is driven by the immune phenotype and that this can be used to select patients for therapy. The small number of patients in the training data also certainly limits this claim (as already noted by the authors).

A few small comments:

It is unclear to what test the p-value shown in Figure 2b corresponds.

It is somewhat surprising, given that this is a radiomics paper, that there are no images of MRIs. Perhaps in Figure 4, the authors could include two example patient images one from G2 and one from G3.

The fonts in Figure 6, mainly the feature names, should be made larger.

Reviewer 2 Report

In their manuscript, Hsu et al. report on characterization of immunophenotypes in glioblastoma. In the age of immunotherapies, this in an important topic to adress.
They use RNAseq data to calculate enrichment of four different immune subsewts in TCGA data and then use the binarized information (high/low) as target for a logistic regression radiomics models after some rounds of feature selection. In a test set they find a significant prognostic value of this radiomics/immunephenotype classifier.
A couple of remarks from my site:

> You state (rightfully) in the introduction that "tumor biopsy may suffer under-sampling when representing the pathological molecular profile in a whole tumor". This is also true for your sutdy: You generalize from the biopsy data to the entire tumor. How do you see this method being used to shed light on heterogeneity (which is a huge problem, as you say)?

> In the test data you have no data on immune enrichment, so you take OS as a surrogate marker (which is fine). However, you can not really validate that you identified an immune phenotype predictive signature (but rather a signature predicting survival).

> Why do you train logReg if you use RF for feature selection / importance? Why not use RF (which is fine both for classification and regression) entirely?

> Why do you predict each immune cell subtype (and not the unsupervised clusters G1..G5) from MRI data? This would mean less (one) model only.

> Why not combine ADC and T1 information?

Reviewer 3 Report

Bo-Kai Hsu, et al. reported a radiomics diagnostic method for the immunophenotyping of glioblastoma by using machine learning analysis of MRI. The results indicate that the patients having tumors enriched with cytotoxic T lymphocytes without infiltration of regulatory T cells (Tregs) and myeloid-derived suppressor cells (MDSC) survived significantly longer than other immunophenotypes. This paper sounds interesting and leading for the future. However, the following points are serious concern.

  1. The number of patients is too small to extract a significant result with the machine-learning method.
  2. Actually, no relationship was confirmed between the radiomics results and the immunophenotyping by the method used here.

The point a) can be overlooked because this paper includes a promising concept for the future. For the point b), the authors should obtain some information about immunological microenvironments for the testing data patients, at least for the population included in the Figure 4. The immunological information needs not to be full but imply some immunological differences, such as CTL immunohistochemistry or so.

Round 2

Reviewer 2 Report

Thank you for your revisions.